# Evolution of a biological thermocouple by adaptation of cytochrome c oxidase in a subterrestrial metazoan, *Halicephalobus mephisto*
Megan N. Guerin[1,4], TreVaughn S. Ellis[1,4], Mark J. Ware[1], Alexandra Manning[1], Ariana A. Coley[1], Ali Amini [2], Adaeze G. Igboanugo[1], Amaya P. Rothrock[1], George Chung [3], Kristin C. Gunsalus [3] & John R. Bracht [1]✉

In this study, we report a biological temperature-sensing electrical regulator in the cytochrome c oxidase of the Devil Worm, *Halicephalobus mephisto*. This extremophile metazoan was isolated 1.3 km underground in a South African goldmine, where it adapted to heat and potentially to hypoxia, making its mitochondrial sequence a likely target of adaptational change. We obtained the complete mitochondrial genome sequence of this organism and show through dN/dS analysis evidence of positive selection in *H. mephisto* cytochrome c oxidase subunits. Seventeen of these positively selected amino acid substitutions were located in proximity to the H- and K-pathway proton channels of the complex. Surprisingly, the *H. mephisto* cytochrome c oxidase completely shuts down at low temperatures (20 °C), leading to a 4.8-fold reduction in the transmembrane proton gradient ($\Delta\Psi_m$) compared to optimal temperature (37 °C). Direct measurement of oxygen consumption found a corresponding 4.6-fold drop at 20 °C compared to 37 °C. Correspondingly, the lifecycle of *H. mephisto* takes four times longer at low temperature than at higher. This elegant evolutionary adaptation creates a finely-tuned mitochondrial temperature sensor, allowing this ectothermic organism to maximize its reproductive success across varying environmental temperatures.

*Halicephalobus mephisto*, or the Devil Worm, is a subterranean nematode isolated from 1.3 kilometers below Earth's surface[1]. Endemic to mine water that is warm (37 °C), hypoxic (0.42–2.3 mg/L dissolved O2,), alkaline (pH 7.9), and methane-rich[1], the nuclear genome displayed expanded novel stress response gene families including heat-shock protein (Hsp70) and avrRpt2 induced gene 1 (AIG1)[2]. Here we report the full mitochondrial genome sequence of *H. mephisto* and report the first analysis of cytochrome c oxidase function in this unique organism.

The mitochondrion is a vital, double-membraned organelle that is most well characterized for its role in ATP production, TCA cycle, *β*-oxidation of fatty acids, and the initiation of apoptosis through the mitochondrial stress response[3]. Mitochondrial dysfunction has been linked to human diseases including neurodegenerative disorders[4], cancer[5], and even autoimmune disease[6]. The human and *Drosophila* mitochondrial genomes encode 13 protein-coding genes, 22 transfer RNAs (tRNAs), and two ribosomal RNAs (rRNAs), while most nematodes (including *H. mephisto*) have reduced their repertoire of protein-coding genes to 12, having lost the ATP8 gene[7].

Studies of mitochondrial evolution and function tend to fall into two main types: either reporting of sequence changes (evolution) without functional characterization (often dN/dS analysis), or evaluations of mitochondrial function within established model systems (without evolution). As examples of the first, tantalizing evolutionary sequence changes have been reported in the mitochondrial genomes of mammals[8], primates[9–11], birds[12–14], reptiles[15,16], bats[17] and fruit flies[18,19], yet in all cases the functional impacts were not tested. On the other hand, with the advent of the Seahorse XF Analyzer, a number of studies report mitochondrial functional data from traditional model organisms such as the nematode *C. elegans*[20,21], zebrafish *Danio rerio*[22], and fruit fly *Drosophila melanogaster*[23].

[1]Biology Department, American University, 4400 Massachusetts Avenue, NW, Washington, DC, 20016, USA. [2]Mathematics and Statistics Department, American University, 4400 Massachusetts Avenue, NW, Washington, DC, 20016, USA. [3]Center for Genomics and Systems Biology and Department of Biology, New York University, New York, NY, 10003, USA. [4]These authors contributed equally: Megan N. Guerin, TreVaughn S. Ellis. ✉e-mail: jbracht@american.edu

Interestingly, two case studies examining mitochondrial function in non-model systems with evolutionary adaptation to heat both identified cytochrome c oxidase as sites of evolutionary change. The first study, an examination of heat-tolerant *Saccharomyces cerevisiae* and its cold-adapted relative, *S. uvarum*, both the hot and cold adaptation led to alterations in the mitochondrial genome and specifically the cytochrome c oxidase 1 (COX1) gene[24]. In the second study, of Atlantic killfish, *Fundulu heteroclitus*, mitochondrial oxygen binding was different in thermally adapted subspecies[25], and the authors hypothesized that cytochrome c oxidase sequences are responsible. However, mitochondrial genome sequences remain uncharacterized in these particular subspecies. Therefore, there is a critical need to combine genetic insight with functional data in non-model organisms, particularly of evolutionary importance. Therefore, we chose to study *H. mephisto* and compare it to *C. elegans* as an outgroup[20]. Our approach was to sequence and evaluate the mitochondrial genomic function of *H. mephisto*, within the intact animal, to identify the impact of evolutionary changes during its adaptation to the extreme subsurface environment.

The mechanisms by which natural selection acts on mitochondrial genomes is poorly understood. Given their asexual reproduction, lack of recombination, and highly polyploid nature, the selection pressures acting on mitochondria are quite different from those involving nuclear genes[26,27]. Genes encoded by the mitochondria, especially COX1, are typically subjected to significant functional constraints due to their roles in electron transport and ATP production, resulting in strong purifying selection[26,28–30]. As discussed above, there is countervailing literature documenting multiple cases of positive selection on COX1 particularly under thermal adaptation. Mitochondrial evolution is also significant beyond thermal adaptation: research on brain evolution in humans has found instances of positive selection on cytochrome c oxidase and other mitochondrial proteins, correlating with increases in brain size[9–11].

Here we show that evolution can extensively remodel the COX1 sequence, particularly the proton channels, in a heat-tolerant metazoan. The resultant mitochondria have become temperature sensors, with a finely-tuned thermal response. Because the modified cytochrome c oxidase generates differential electrical voltages (proton gradients) directly linked to energy metabolism, the evolved system functions as a naturally-evolved thermocouple device modulating the life cycle of *H. mephisto*.

## Results

PacBio long-read sequencing of the *H. mephisto* mitochondrial genome produced a 14,349 bp sequence that was 81% AT-rich, including 12 protein-coding genes, 22 tRNAs, and two rRNAs, that agreed closely with those of other nematodes including those of the *Halicephalobus* genus (Fig. 1). Along with most nematodes, *H. mephisto* has lost the atp8 gene, and all genes are transcribed in the same direction, a characteristic of Chromadorea[7]. The control region is 863 bp long, located between tRNA-Isoleucine and tRNA-Arginine, and a remarkable 95% AT (Fig. 1). Of mitochondrial genomes sequenced to date, *H. mephisto* encodes one of the most AT-rich.

To broaden our analysis we included mitochondrial sequences of two additional *Halicephalobus* species: *H. gingivalis*, a horse parasite[31], and species NKZ332, originally isolated by Natsumi Kanzaki in association with the Japanese termite, *Reticulitermes speratus* and sequenced by Erik Ragsdale[32]. Both the nuclear and mitochondrial genomes of this organism show distinct molecular markers from all known *Halicephalobus* species, and in phylogenetic analysis this species is a well resolved sister group to both *H. mephisto* and *H. gingivalis*. Therefore we have renamed strain NKZ332 *Halicephalobus consperatus* (for 'with speratus') after its association with the termite. Unfortunately the nature of the interaction, whether phoretic, symbiotic, or parasitic, may never be resolved because the species is no longer in culture (E. Ragsdale, personal communication). Overall, for the present study we evaluated 36 nematodes and two outgroups, *Homo sapiens* and *Drosophila melanogaster*, for 38 total mitochondrial genomes (accession numbers in Supplemental Table 1).

Utilizing the 12 protein-coding genes of these 38 mitochondrial genomes we constructed a concatenated mitochondrial protein Maxmimum Likelihood phylogeny (Fig. 2a) which is broadly consistent with other nematode mitochondrial gene trees[33]. In particular, the monophyly of classes Enoplia (Clade II) and Chromadorea (containing Clades III, IV, and V) are cleanly recovered; however within Chromadorea both clades III (Spirurina) and IV (Tylenchina) are not monophyletic, consistent with previous mitochondrial phylogenies[7,33]. Clade V (Rhabditina) is monophyletic with the exception of two newly placed *Diploscapter* species (*D. pachys* and *D. coronatus*) emerging as sister clades to *Bursaphelenchus xylophilus* (Clade IV) (Fig. 2a). Given that Maximum Likelihood analysis may suffer from long-branch attraction artifacts[34] we re-analyzed our data by Bayesian methods, recovering a nearly identical tree again with both *Diploscapter* species emerging as sisters to *B. xylophilus* (Fig. S1). Given the abundance of nuclear genomic data demonstrating that both *Diploscapter* species are closer relatives of *C. elegans* and members of Rhabditina[35,36], not Tylenchina, we conclude that the mitochondrial genomes of these species exhibit significant homoplasy, but we have not explored this phenomenon further.

To evaluate potential selective events in the *Halicephalobus* lineage we performed a likelihood ratio test (LRT) of positive selection using PAML[37], which requires the phylogeny to accurately capture the species' evolutionary history. Therefore, we manually re-located the two *Diploscapter* species to their correct position (based on nuclear phylogenetics) within Rhabditina as indicated by a red asterisk on Fig. 2a. Supporting the correctness of this move, the PAML-derived log-likelihood (lnL) values increased by an average of 23.2 across the branches shown in Fig. 2b after relocating *Diploscapter* (max, 25.2 and min, 22.3). We tested COX1, 2, and 3 and ND4 using the branch-sites test for positive selection (also known as Model A), generating Bonferroni-corrected *p* values for each branch[38]. While we had expected positive selection might be detectable along the most recent lineage leading to *H. mephisto* (branch A), we instead found statistically robust positive selection along three ancestral branches B, C, and D (Fig. 2b). COX1 is a particular focus of evolutionary innovation as the only gene with positive selection across three of four branches tested (Fig. 2b).

The division of branches into A, B, C, and D divides the evolutionary selection into distinct times: the divergence of the suborder Tylenchina (branch D); the diversification of family *Panagrolaimidae* (branch C); the common *Halicephalobus* ancestor (branch B); finally, the lineage (branch A) leading to *H. mephisto*. We therefore analyzed amino acid substitution patterns in COX1 along these lineages. To do this, we examined fixed derived non-synonymous mutations (FdNs) shared within the clades[39] (Table 1). We identified 18 amino acid substitutions in COX1, one in COX2, and one in COX3 by manual inspection of sequence alignments, for 20 total substitutions (Table 1). In each case a single nonsynonymous single nucleotide polymorphism (snp) was responsible for the substitution and was conserved within the clade; sometimes with adjacent synonymous sequence changes preserving the altered amino acid (Table 1). Given the wide range of evolutionary divergence between species within Tylenchina, *Panagrolamidae*, and even within *Halicephalobus*, this pattern of derived amino acid substitutions suggests a functional preservation and is consistent with positive selection[39].

PAML also reports sites of selection as part of its output. These sites we interpret with caution given the high divergence between sequences in our phylogeny raising concerns around synonymous site (dS) saturation noted in some tests of positive selection[40]. The PAML sites corroborated the alignment-based FdNs sites but also identifies other sites. For example, for COX1, PAML identified 3 sites on branch B (1 is an FdNs); 14 sites on branch C (9 FdNs); and 12 sites on branch D (8 are FdNs). Thus these PAML-identified sites are confirmatory of the FdNs identified through sequence alignment. We also note that in general the branch-sites test (which we implemented) has been found to be conservative under synonymous site saturation, displaying a loss of power rather than false positive inferences, at saturation of dS[40]. Overall, the statistically robust inference of

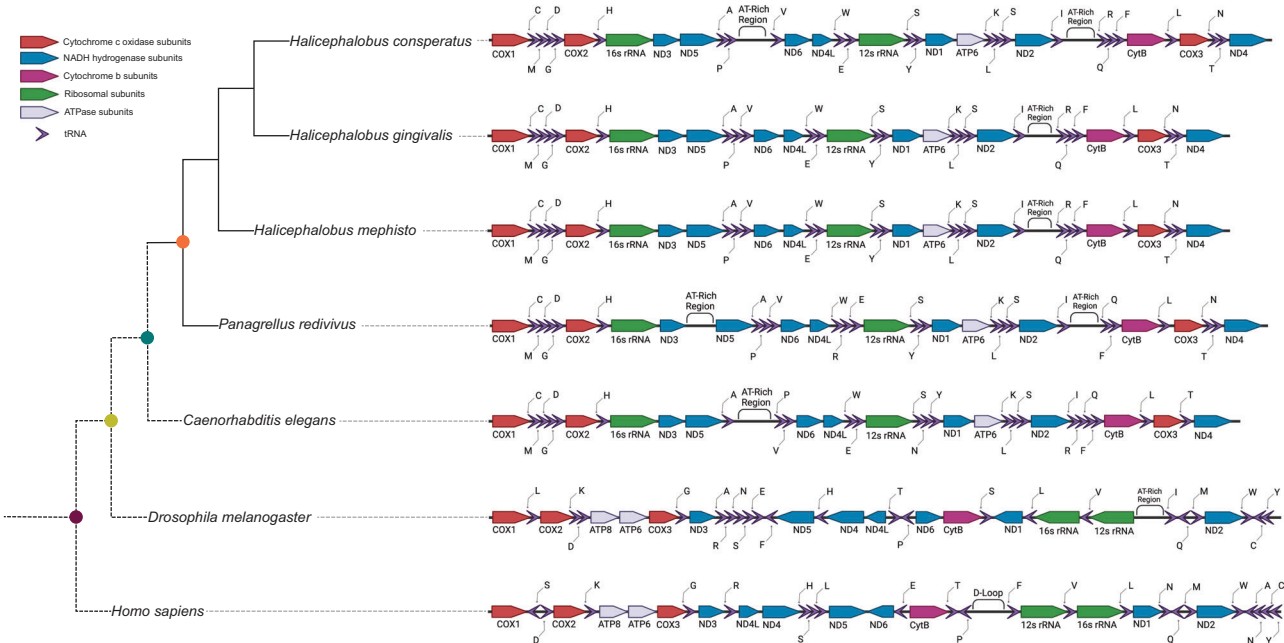

**Fig. 1 | Overview of mitochondrial architectures in human, *Drosophila*, and nematodes.** Image created with BioRender.com. Note, *Halicephalobus consperatus* was formerly *Halicephalobus* strain NKZ332. Transfer RNAs indicated by their corresponding single-letter amino acid.

positive selection by LRT and 20 FdNs amino acid substitutions warrant further investigation into their function.

We hypothesized that the 20 amino acid substitutions might co-localize in the cytochrome c oxidase structure. To evaluate this, we created a 3D homology model of the COX1, -2, and -3 proteins of *H. mephisto* by SWISS-MODEL based on the well-resolved crystal structure of bovine cytochrome c oxidase (3abm in PDB; 1.95 Å resolution). To independently assess the accuracy of this model, we also constructed AlphaFold structure predictions for COX1, -2, and -3, which were nearly superimposable with the SWISS-MODEL structure: RMSD of 0.565 Å for COX1, 1.592 Å for COX2 and 0.815 Å for COX3 (Supplemental Fig. 2). Because the SWISS-MODEL structure situates all three subunits and is nearly identical to the separately predicted AlphaFold structures, we used it for illustrative purposes, for locating the *H. mephisto* amino acid changes in space.

We therefore mapped the identified FdNs amino acid changes onto the SWISS-MODEL structure. Of the 18 located in COX1, 17 (94%) are proximal to (within 13 Å) proton channels: the H-pathway or the K-pathway (Fig. 3c, d). Six FdNs are within four Å of a proton channel (Fig. 3c, d, Table 1). Due to the separation of evolutionary timeframes by branch, we can see that channel modification occurred in stages, first along the H-pathway (branch D) then along the K-pathway (branch C) and finally back along the H-pathway again (Branch B) (Fig. 3c, d).

COX2 and -3 showed a single FdNs each, consistent with their much smaller sizes relative to COX1. For COX2, the substitution occurs in a loop near the cytochrome c binding site, while for COX3, the substitution is located on the mitochondrial matrix side of the mitochondrial membrane (Fig. 3a). The possible functional impact of these alterations is not clear from these data, but the COX3 substitution is very close to the interface with the COX6A2 subunit interface.

Considering that cytochrome c oxidase serves as the locus for oxygen reduction, we hypothesized that the observed amino acid substitutions might confer an adaptive benefit under hypoxic conditions, particularly because *H. mephisto* was initially isolated from hypoxic subterranean water[1]. To test this, we performed anaerobic chamber culture using oxygen absorbing sachets at 20 °C and 37 °C, producing an environment of less than 0.1% oxygen. For both *H. mephisto* and *C. elegans* synchronized L1 hatchling worms were cultured on standard OP50 food on NGM plates. We found that both *C. elegans* and *H. mephisto* survived 9 days of severe hypoxia

at 20 °C (Fig. 4). However by day 22 only *C. elegans* had survivors and by day 29 all worms had perished in the hypoxic environment (Fig. 4). At 37 °C *H. mephisto* were markedly hypoxia intolerant, exhibiting 100% lethality by day two (Fig. 4). These findings collectively suggest that *H. mephisto* lacks adaptation to severe hypoxia, indicating that postive selection on cyto-chrome c oxidase is more likely attributable to thermal adaptation.

Given the potentially function-altering changes in the proton trans-location H- and K-pathways (Fig. 3c, d) we performed analyses of the mitochondrial proton gradients using tetramethylrhodamine, ethyl ester (TMRE) (Fig. 5a–e). This cationic dye is imported into the inner lumen of the mitochondria proportionally to the proton gradient across the inner mitochondrial membrane, $\Delta\Psi_m$. The proton gradient is the product of three proton pumps: complex I, III, and IV[41] (Fig. 5f). To isolate the effect of complex IV alone, we used 25 mM sodium azide, a specific inhibitor of cytochrome c oxidase (Complex IV), which leaves Complexes I and III unaffected[42] (Fig. 5g). Thus, by comparing untreated with azide-treated nematodes, we can directly assess the contribution of Complex IV activity to $\Delta\Psi_m$. Under exposure to sodium azide, it is assumed that fumarate acts as the terminal electron acceptor instead of $O_2$, similar to what occurs in mammalian hypoxia[43] (Fig. 5g).

Using this TMRE-based imaging method, we found that both *H. mephisto* and *C. elegans* display similar $\Delta\Psi_m$ (p = 0.13, two-way ANOVA and Tukey's HSD post-hoc test) when comparing their optimum tem-peratures: 20 °C for *C. elegans* and 37 °C for *H. mephisto* (matching its native subterranean habitat[1]). For *C. elegans*, at 25 °C (near its thermal maximum temperature of 26 °C), a significant reduction in $\Delta\Psi_m$ is apparent, an apparent decoupling of the proton gradient due to thermal stress[44]. At 20 °C, Complex IV is inactive in *H. mephisto* since sodium azide treatment shows no change in $\Delta\Psi_m$ (p = 0.99, ANOVA and Tukey's HSD post-hoc test, Fig. 5h). These results suggest that, for *H. mephisto*, the proton pumping at 20 °C is conducted largely by complexes I and III leading to a very strong reduction in the proton gradient, and thus a lower voltage, across the membrane. We observed that $\Delta\Psi_m$ changes by 4.8-fold (Fig. 5h) between 37 °C and 20 °C. This difference is directly due to altered cytochrome c oxidase activity: at 37 °C the inhibition of cytochrome c oxidase by sodium azide led to a 5.1-fold drop in proton pumping while it only caused a 0.98-fold drop at 20 °C because of the inactivity of cytochrome c oxidase at this tempera-ture (Fig. 5h).

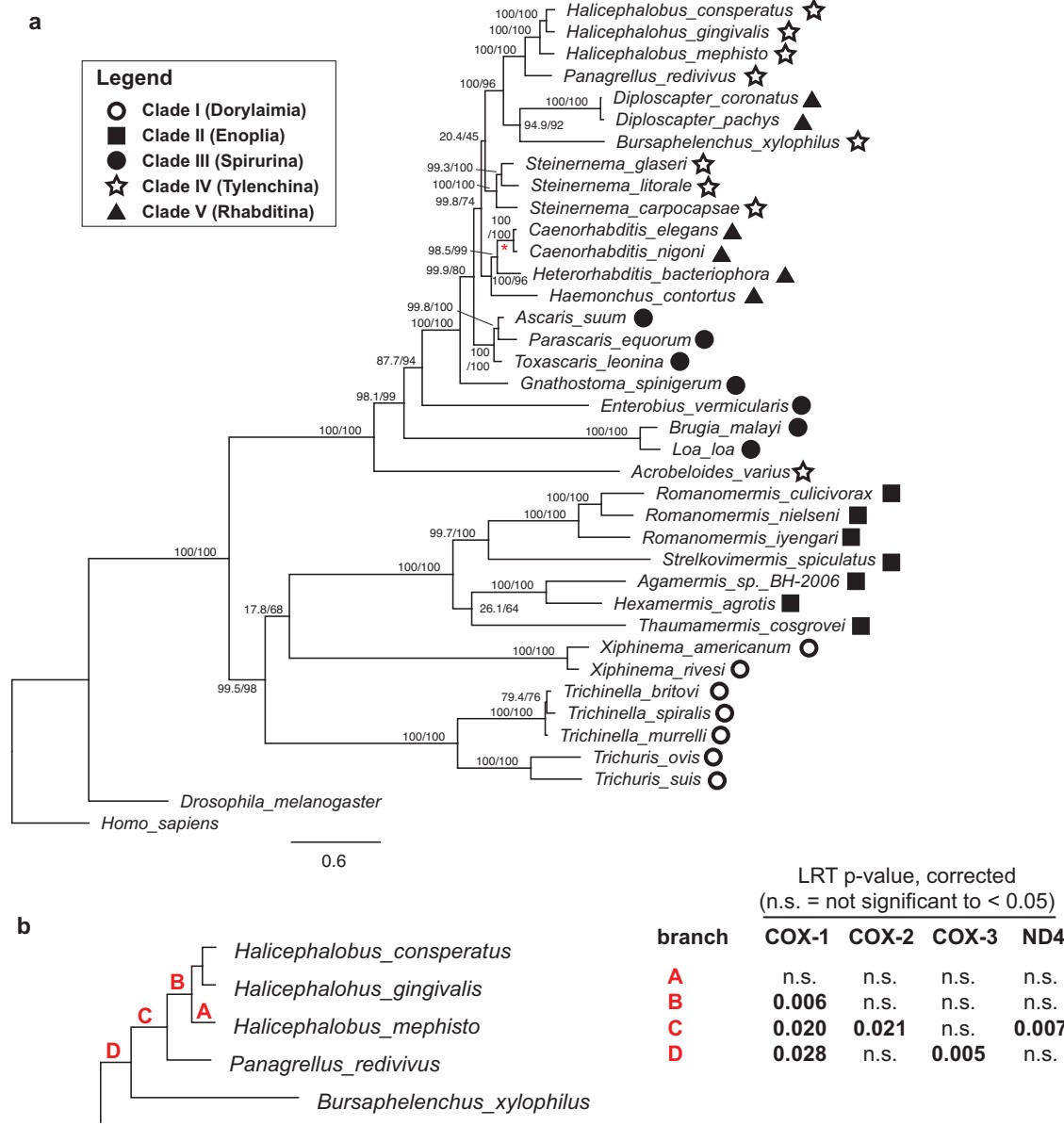

**Fig. 2 | Phylogeny and dN/dS analysis. a** Combined mitochondrial ML phylogeny of nematodes created with IQtree and model mtZOA+F+I+G4. Bootstrap values indicate SH-aLRT support (%) / ultrafast bootstrap support (%). Scale bar indicates substitutions per site. **b** Branch labels and Paml analysis by branch-sites. In the table the *p* values for the Likehood Ratio Tests (LRTs) for positive selection are shown. For each branch PAML was run twice, once to allow an estimated dN/dS ($\omega_2$) for a subset of sites on the foreground branch; once to fix $\omega_2$ at 1 for the foreground branch (in all cases another $\omega_1$, measures purifying selection of most sites). By taking the ratio of maximum likelihood values, a *p* value is measured using a Chi-Square table with 2 degrees of freedom (critical values are 5.991 for *p* value < 0.05 and 9.210, *p* value < 0.01). Bonferroni correction was performed to correct for multiple hypothesis testing.

The temperature-controlled difference in the proton pumping by cytochrome c oxidase could be due to amino acid changes or changes in gene expression. To test this, we examined temperature-dependent mitochondrial gene expression from our previously published RNAseq dataset[2]. For COX1, 2, and 3, temperature made only a slight difference in expression (Fig. 5i). COX1 was elevated 1.8-fold at 38–40 °C relative to 25 °C while COX2 and COX3 were essentially unchanged, 1.1 and 0.98-fold different, respectively (Fig. 5i). Analysis of all 12 protein-coding mitochondrial sequences revealed that all were regulated by less than 2-fold by the temperature changes (Fig. S3). Given that $\Delta\Psi_m$ changes by 4.8-fold at 20 °C relative to 37 °C ($\Delta\Psi_m$ at 20 °C and 25 °C are indistinguishable, Fig. 5h) we conclude that the dramatic effect of high vs. low temperature on $\Delta\Psi_m$ cannot be explained by transcriptional regulation alone and most likely reflects a difference in protein function, not regulation.

Mitochondrial respiration couples electron transport to its acceptor, oxygen, with the production of a proton gradient ($\Delta\Psi_m$) and ATP. Therefore, we directly assayed the mitochondrial oxygen consumption rate with a Seahorse HS Mini analyzer[20,21] comparing mitochondrial oxygen consumption in *H. mephisto* at 20 °C, 25 °C, 37 °C and 40 °C (Fig. 6a) and *C. elegans* at 20 °C and 25 °C (Fig. 6b). In *H. mephisto*, we observed that basal respiration, maximal respiration, and spare respiratory capacity all increase from 20 °C to 25 °C, remain high at 37 °C, and decrease at 40 °C, which is the maximum survivable temperature for *H. mephisto* in the laboratory. Indeed, we found that while we could shift the worms from 37 °C to 40 °C and they survive (see Methods), the population did not grow at this temperature. This suggests that the organisms cannot reproduce at this thermal extreme. Worms subjected to 41 °C died overnight.

## Table 1 | Positively Selected amino acids of COX1, 2, and 3

| B. taurus coordinates | H. sapiens | B. taurus | C. elegans | D. pachys | D. coronatus | B. xylophilus | P. redivivus | H. mephisto | H. gingivalis | H. consperatus | H. mephisto coordinates | H-pathway proximal | K-pathway proximal | chemical change high | notes | synonymous substitutions observed? |
|---|---|---|---|---|---|---|---|---|---|---|---|---|---|---|---|---|
| **COX1-Branch D** | | | | | | | | | | | | | | | | |
| 29 | L | V | V | V | V | L | L | L | L | L | 36 | * | | Low | Nonpolar-nonpolar | yes |
| 35 | L | L | L | N | N | M | M | M | M | M | 42 | ** | | Low | Nonpolar - Nonpolar | yes |
| 39 | A | A | L | L | L | F | F | F | F | F | 46 | ** | | Low | Nonpolar-nonpolar | yes |
| 46 | N | T | F | F | F | Y | Y | Y | Y | Y | 53 | * | | High | Nonpolar-polar | no |
| 215 | L | L | L | L | L | I | I | I | I | I | 221 | | | Low | Nonpolar-nonpolar | no |
| 462 | L | L | T | N | N | V | V | V | V | V | 468 | ** | | High | Polar-nonpolar | yes |
| 464 | A | A | G | G | G | S | S | S | S | S | 470 | ** | | High | Nonpolar-polar | yes |
| 479 | K | K | Y | Y | Y | F | F | F | F | F | 485 | * | | High | Polar-nonpolar | yes |
| **COX1-Branch C** | | | | | | | | | | | | | | | | |
| 89 | A | A | A | A | A | A | S | S | S | S | 96 | | * | High | Nonpolar-polar | no |
| 259 | T | T | L | I | I | L | M | M | M | M | 265 | | ** | Low | Nonpolar-nonpolar | no |
| 270 | Y | Y | A | S | S | S | T | T | T | T | 276 | | * | Low | Polar-polar | no |
| 353 | I | L | L | L | L | L | I | I | I | I | 359 | | * | Low | Nonpolar-nonpolar | no |
| 399 | L | L | F | F | F | L | Y | Y | Y | Y | 405 | * | * | High | Nonpolar-polar | no |
| 403 | Y | Y | Y | V | V | F | L | L | L | L | 409 | * | | Low | Nonpolar-nonpolar | no |
| 441 | S | S | L | L | L | L | M | M | M | M | 447 | * | | Low | Nonpolar-nonpolar | yes |
| 494 | W | W | Y | V | V | N | G | G | G | G | 500 | | ** | medium | Polar-nonpolar | yes |
| 495 | L | L | C | V | A | M | S | S | S | S | 501 | * | * | High | Nonpolar-polar | no |
| **COX1-Branch B** | | | | | | | | | | | | | | | | |
| 476 | F | F | F | F | F | F | F | M | M | M | 482 | * | | Low | Nonpolar-nonpolar | yes |
| **COX2-Branch C** | | | | | | | | | | | | | | | | |
| 94 | S | S | N | F | Y | N | Q | Q | Q | Q | 97 | | | Low | | no |
| **COX3-Branch D** | | | | | | | | | | | | | | | | |
| 156 | R | R | R | S | S | D | D | D | D | D | 153 | | | High | Positive or polar to negative | no |

**Within 4 angstroms of channel.
*Within 13 angstroms of channel.

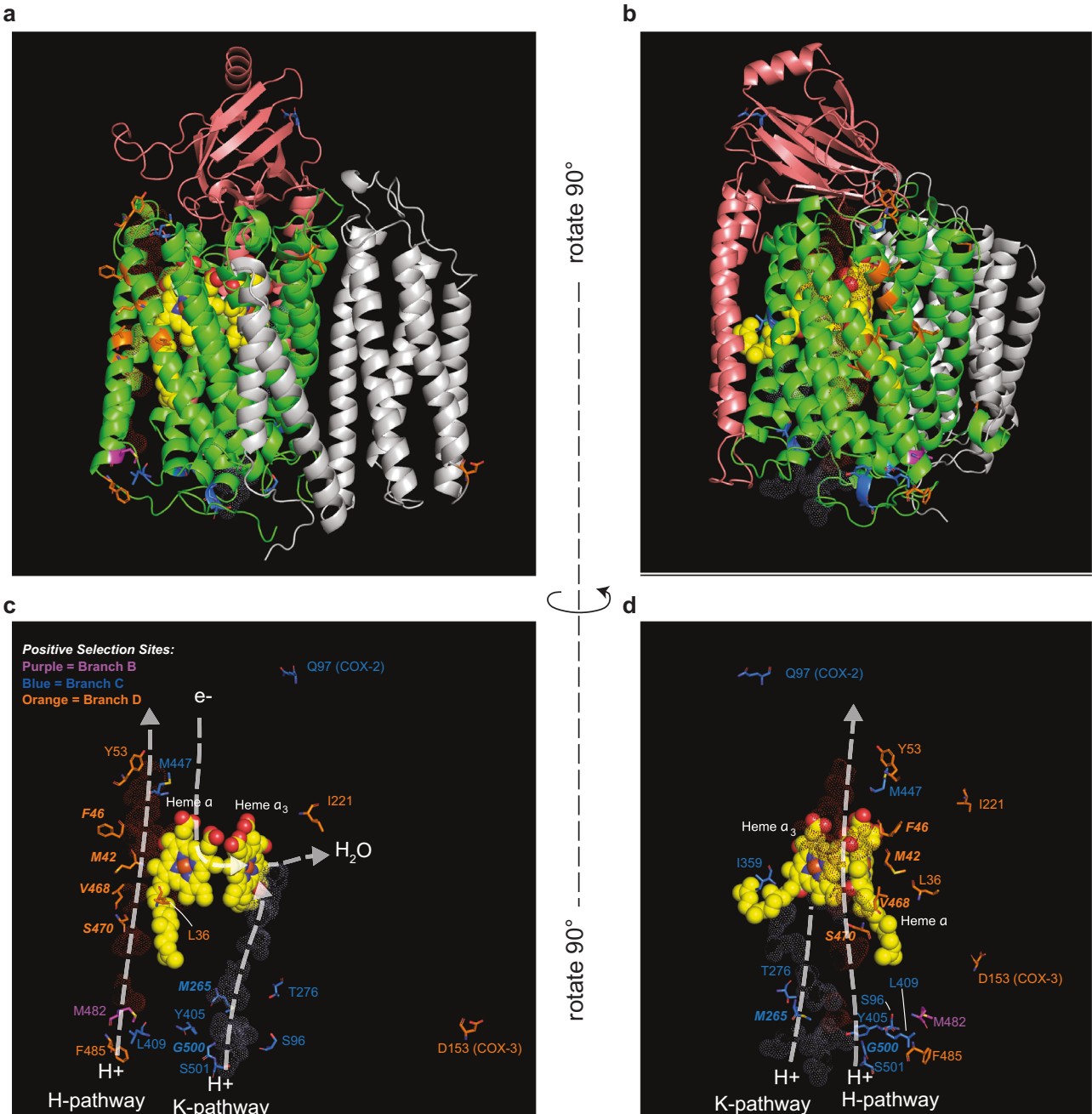

**Fig. 3 | Homology model of *H. mephisto* COX1, COX2, and COX3 modeled on bovine crystal structure 3abm. a** Ribbon structure showing the three *H. mephisto* proteins COX1 = green; COX2= salmon pink; and COX3=light gray. Heme groups from bovine crystal structure in yellow. **b** Same as A, but rotated 90 degrees. **c** Cutaway view showing only the heme and positively selected sites with their residue numbers. For residues within 4 Å of their channel are in bold italic font while those within 13 Å are in regular font. The H- and K- proton pathways are shown as red or light blue dots outlining the residues from the bovine structure corresponding to these pathways. **d** Same as C but rotated 90 degrees.

Contrary to a previous report of a "flat" thermal response in *C. elegans*[20] we observed a 2.5-fold increase in oxygen consumption of *C. elegans* at 25 °C relative to 20 °C (Fig. 6d). In *H. mephisto's* oxygen consumption rate is decreased by 4.6-fold at 20 °C (minimal) compared to 37 °C (maximal) (Fig. 6c), consistent with the proton gradient data, which showed a 4.8-fold difference (discussed above and Fig. 5h). Unexpectedly, *H. mephisto's* basal, spare, and maximal respiration rates at 25 °C were statistically indistinguishable from 37 °C (Fig. 6c), contrasting with the proton data, which indicated low cytochrome c oxidase activity at 25 °C (Fig. 5h). This could suggest proton gradient uncoupling at 25 °C.

However, we interpret this finding with some caution because DMSO clearly affected respiration more strongly at higher temperatures (Fig. S4). Because our basal respiration calculation includes DMSO, this interaction of DMSO with temperature may be a confounding factor. Supporting this, we note that the untreated 25 °C oxygen consumption rate prior to drug injection 1 is quite similar to 20 °C, and not similar to the rate at 37 °C (Fig. 6a). It is, however, notable that maximal respiration at 25 °C matches quite closely with that of 37 °C, while at 20 °C it does not (Fig. 6a, c), hinting at unexpected metabolic differences between animals exposed to these temperatures.

**Fig. 4 | Survival of *H. mephisto* and *C. elegans* in hypoxic conditions at different temperatures.** Plotted are mean and standard deviations across at least *n* = 3 experimental replicates.

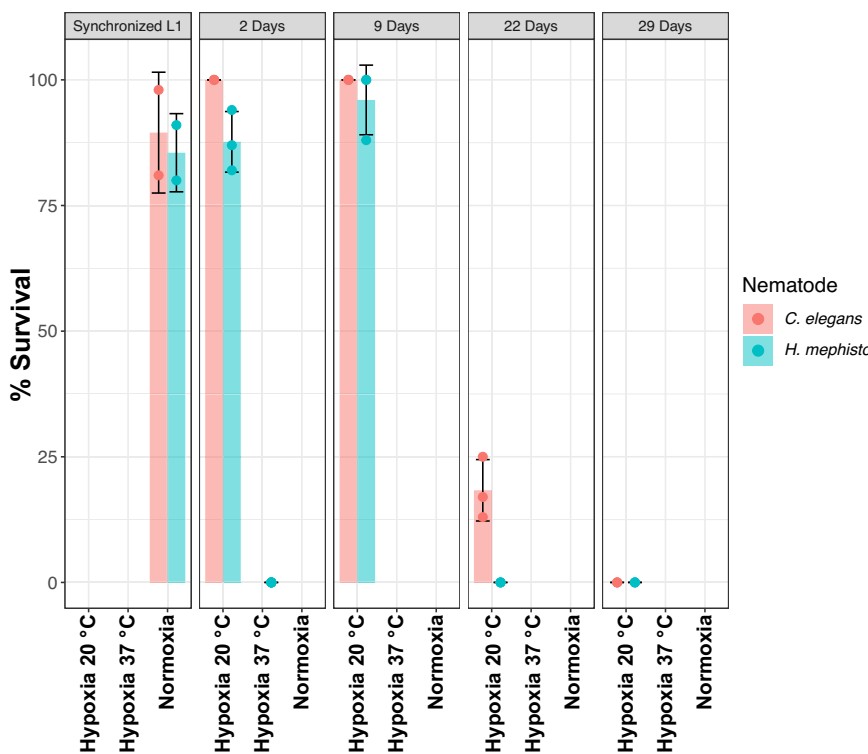

Temperature appears to play a central role in regulating *H. mephisto* metabolism. Given that at 20 °C relative to 37 °C oxygen consumption was reduced 4.6-fold and mitochondrial proton gradient decreases 4.8-fold we would predict that these organisms experience decreased ATP production[45] and lower metabolic rates. To test this, we measured the time per lifecycle (L1 to L1) at 20 °C and 37 °C. We found a statistically significant 4-fold increase in time per cycle at 20 °C (mean 195.2 h) relative to 37 °C (mean 50.5 h) (Fig. 6e).

## Discussion

Our analysis of COX1, COX2, and COX3 uncovered a striking evolutionary pattern of fixed, derived nonsynonymous substitutions (FdNs) along the H- and K-pathway proton translocation channels (Fig. 3c, d). These substitutions are extremely likely to alter the function of cytochrome c oxidase, and several of the sites with substitutions have been identified in other systems as either critical or adjacent to critical residues in proton translocation. The H-pathway transports protons from the mitochondrial lumen through a water accessible channel that leads to heme a, and protons from there move through a hydrogen bond network to be ejected into the intermembrane space[46]. A peptide bond between S441 and Y440 transfers the proton, and a substitution S441P abolishes the pumping of the proton in HeLa cells[47]. S441 in cow corresponds to *H. mephisto* M447, one of the FdNs from Branch C, making this a particularly intriguing substitution potentially impacting proton pumping.

One FdNs, F46 (cow A39) on Branch D, is immediately adjacent (in primary sequence) to a critical residue: bovine and *H. mephisto* R38[48]. R38 acts on the opposite end of the hydrogen-bond network from cow S441/ *H. mephisto* M447, serving to feed protons from the heme a into the hydrogen-bond network[46,48]. (The R38 residue is completely conserved in all species we analyzed). There are 7 FdNs along the water channel leading up to the heme a (F485, L409, M482, S470, L36, V468, M42) and Y53 is near the ejection site in the intermembrane space (Fig. 3c, d). Thus, the conserved nonsynonymous changes along the H-pathway go all the way from the water channel at the mitochondrial lumen, to the heme and hydrogen bond network, and to the proton ejection site near the inter-membrane space[46–48].

Several substitutions cause significant chemical alterations. Within the H-pathway channel, there are seven substitutions: three transition from nonpolar to polar and two switch from polar to nonpolar (see Table 1). On branch D, four of eight substitutions make a polar to nonpolar change or vice versa. The nine substitutions along branch C are more conservative with only three nonpolar to polar changes (Table 1).

Mitochondrially encoded proteins are traditionally found to be under strong purifying selection and high functional constraint[26,28–30]. Supporting this, deleterious mutations are rapidly eliminated in oocytes[49] of mice[50], worms[51] and flies[52], and improved quality control of somatic deleterious mitochondrial mutations was able to extend the lifespan of flies[53]. Despite this strongly purifying pressure, notable examples of positive selection in mitochondrial genes are known, including mammals[8], primates[9–11], bats[17], birds[12–14], turtles[16], snakes[15], and fruit fly[18,19].

The study of snakes found 23 amino acid sites under positive selection in COX1[15]. Of these 23 amino acid sites, three overlap with those from our study: cow / *H. mephisto* L35/M42, A89 / S96, and L353/I359. Of these, only one has the same substitution in *H. mephisto* and snakes: cow L35/M42 (Table 1) also occurs in a blind snake species *Amerotyphlops reticulatus*[15]. The other two sites, A89/S96 and L353/I359 are not convergently mutated in snakes and nematodes. The other studies showing mitochondrial positive selection focus either on different components of the electron transport chain entirely or identify cytochrome c oxidase substitutions not shared in our study.

Through PAML LRT analysis we uncovered statistical evidence of episodic positive selection in the life-history of *H. mephisto* particularly for the COX1 gene (Fig. 2b). Because *H. mephisto* did not show tolerance to hypoxia (Fig. 4), our data suggest that the positive selection signature we identified was driven by adaptation to temperature rather than conditions of low oxygen. Notably, exposing *H. mephisto* to both high temperatures and severe hypoxia led to total lethality within just two days (Fig. 4). This synthetic lethality is perhaps not unexpected since both hypoxia and elevated temperature are known to enhance the generation of reactive oxygen species (ROS) in human cell cultures and various other organisms[44,54,55].

Our data show that *H. mephisto* metabolism is exquisitely tuned to temperature. The mitochondrial membrane proton gradient $\Delta\Psi_m$ (Fig. 5h)

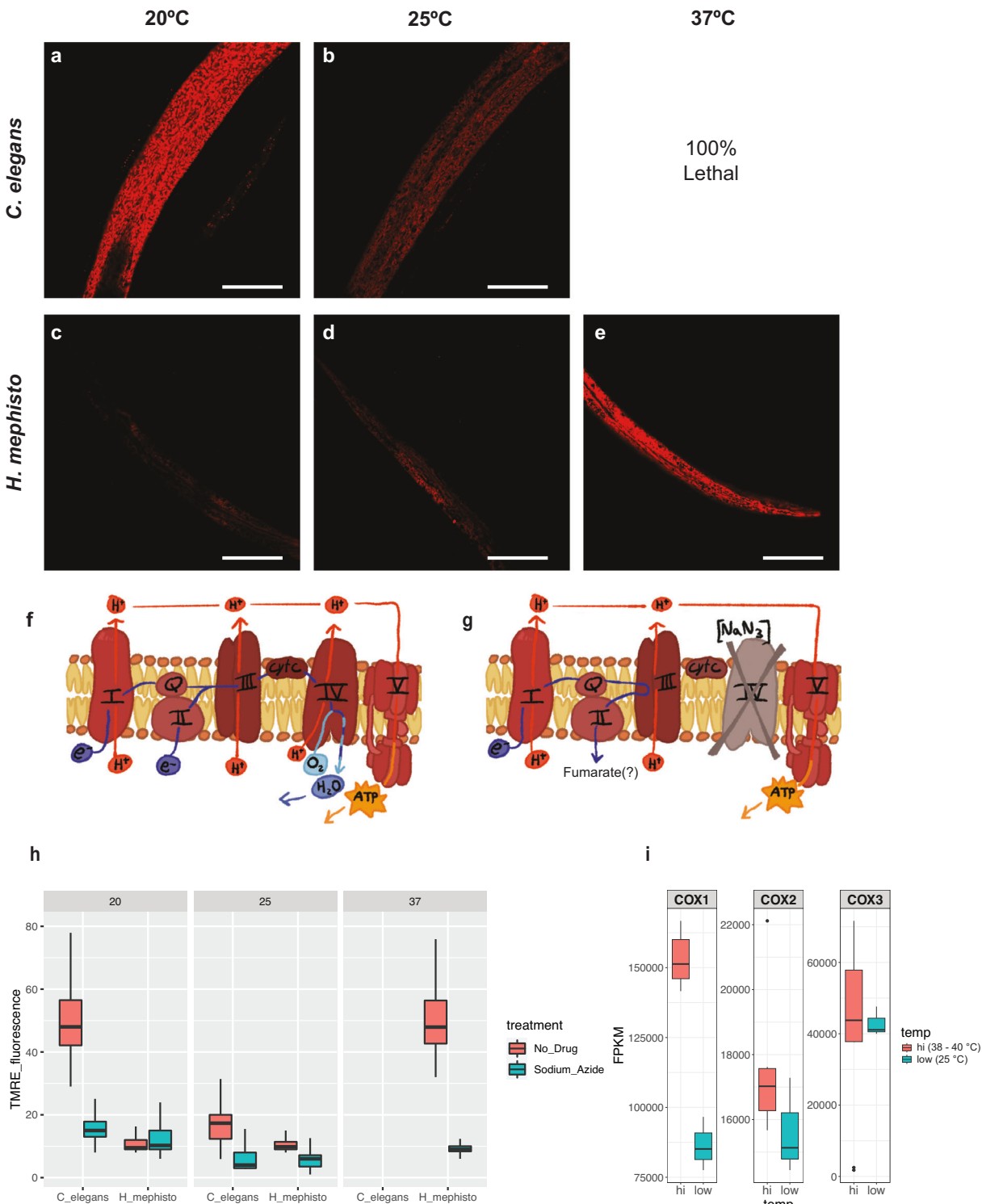

**Fig. 5 | Quantification of mitochondrial proton pumping activity in *H. mephisto* and *C. elegans*.** Representative TMRE images from **a** *C. elegans* at 20 °C. **b** *C. elegans* at 25 °C. **c** *H. mephisto* at 20 °C. **d** *H. mephisto* at 25 °C. **e** *H. mephisto* at 37 °C. **f** Schematic of the electron transport chain showing proton pump and electron flow. **g** Schematic of the electron transport chain illustrating the effect of sodium azide, which blocks Complex IV while leaving remaining proton pumping Complexes I and III unaffected. **h** Boxplot showing the relative TMRE signal from *C. elegans* or *H. mephisto* at different temperatures with or without sodium azide. All boxes in the plot are statistically different from each other, by ANOVA and Tukey's HSD post-hoc test to $p < 0.001$, except for the 20 °C *H. mephisto* no-drug vs. sodium azide treated

($p = 0.99$), *H. mephisto* 37 °C no-drug vs. *C. elegans* 20 °C no-drug ($p = 0.13$), and 25 °C *H. mephisto* sodium azide vs. *C. elegans* 25°C sodium azide ($p = 0.39$). In total, $n = 6119$ and $n = 22,462$ individual mitochondria were assessed for *C. elegans* at 20 °C and 25 °C respectively, and $n = 18,472$ and $n = 2039$ mitochondria for *H. mephisto* at 25 °C and 37 °C respectively. **i,** RNA-seq analysis of COX1, 2, and 3 expression under different temperatures, with $n = 3$ experimental replicates for low temperature (20 °C) and $n = 6$ experimental replicates for high temperature (38–40 °C). FPKM, Fragments per Kb per Million mapped reads. TMRE, TetraMethylRhodamine, Ethyl ester, perchlorate. All scale bars in worm images represent 50 $\mu$m.

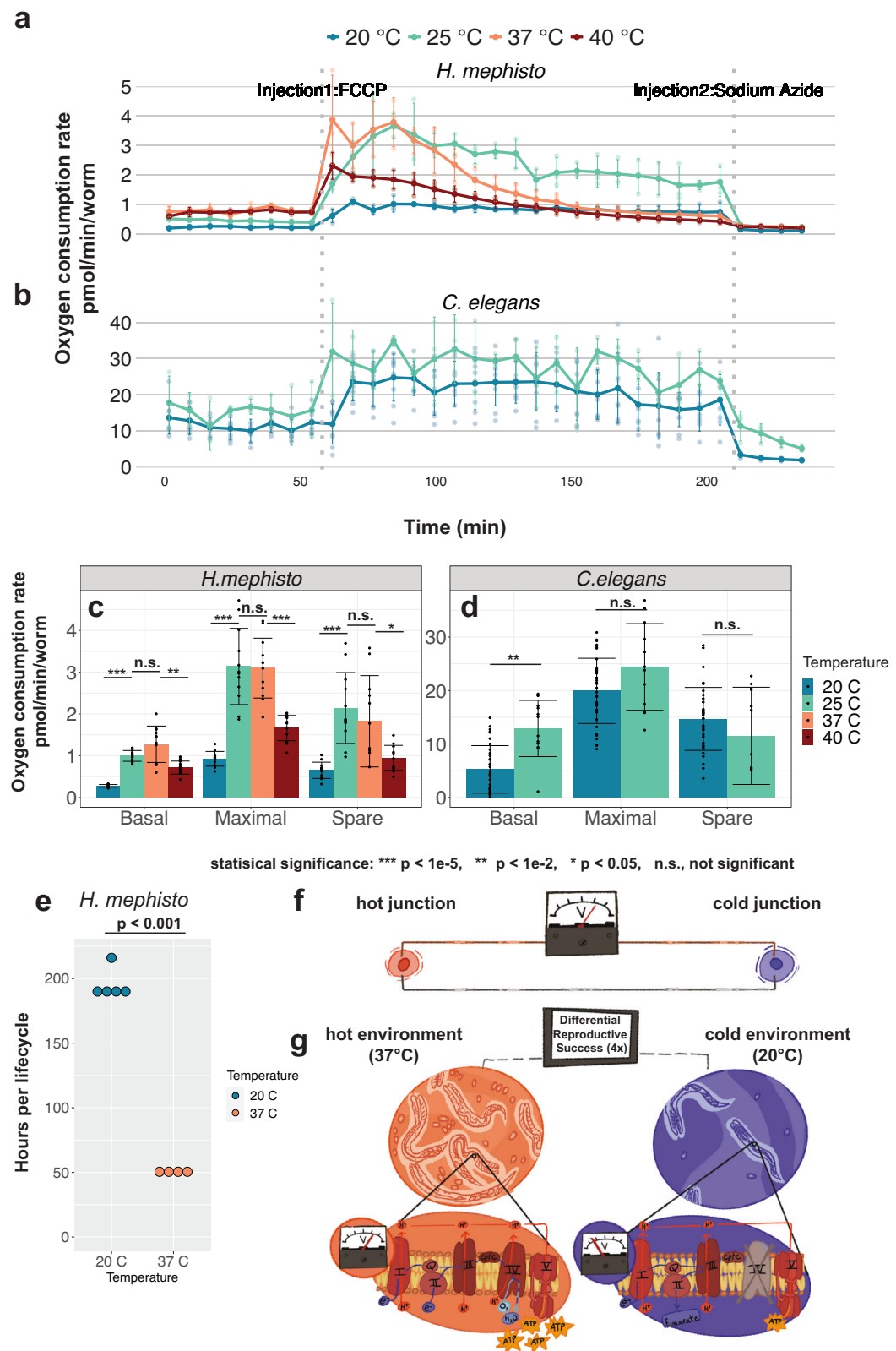

statisical significance: *** p < 1e-5,   **  p < 1e-2,   * p < 0.05,   n.s., not significant

and oxygen consumption rate (Fig. 6c) both decrease by near-identical amounts at lower temperature (4.8-fold and 4.6-fold) while lifecycle time increases by 4-fold (Fig. 6g). Because lifecycle rate extrapolates to population growth rate[56], *H. mephisto* cultivated at lower temperatures will grow more slowly as a population, while growing much more rapidly at the optimal (higher temperature) thermal environment. Together these data elegantly

link environmental temperature, mitochondrial respiration, and the population growth rate.

Our data do not directly prove that positively selected residues drive the observed thermal tuning. However, several lines of evidence support a central role for the positively selected amino acid substitutions. We show that *H. mephisto* is not well-adapted to severe hypoxia (Fig. 4) suggesting

**Fig. 6 | Direct measurement of oxygen consumption rates by Seahorse XF Mini.**
**a** Oxygen consumption rate (OCR) for *H. mephisto* at 20 °C (blue), 25 °C (green), 37 °C (orange), and 40 °C (red). Shown is mean with standard deviation of 32-point time course with 40 $\mu$M FCCP and 25mM sodium azide added at the labeled dotted gray lines. **b** Oxygen consumption rate (OCR) for *C. elegans* at 20 °C (blue) and 25 °C (green), plotted as in panel (a) with mean and standard deviation of 32-point time course. **c** Basal respiration (at 1% DMSO), maximal capacity, and spare capacity for *H. mephisto* at 20 °C, 25 °C, 37 °C and 40 °C. Shown is the mean and standard deviation. Statistical significance assessed by pairwise comparisons using Wilcoxon rank sum exact test. **d** Basal respiration (at 1% DMSO), maximal capacity, and spare capacity for *C. elegans* at 20 °C and 25 °C. Shown is the mean and standard deviation, with statistical significance calculated using one-way ANOVA and Tukey's HSD

post-hoc test. For all Seahorse experiments, $n = 3$ independent replicate wells per temperature were measured, with the exception of *C. elegans* at 20 °C where $n = 9$ independent replicates were assessed. **e** Time required (in hours) to complete one life cycle in *H. mephisto* at 20 °C and 37 °C. Shown are mean and standard deviation. Statistical significance assessed by Student's two-tailed T-test for samples of unequal variance; $n = 5$ and $n = 4$ experimental replicates were assessed for 20 °C and 37 °C, respectively. **f** Schematic of an engineered thermocouple device, which senses temperature as an electrical differential (voltage) between two junctions (hot and cold). **g** Schematic of a biological thermocouple, in which varying environmental temperatures translate into both voltage differences (within the mitochondrial inner membranes) and differential reproductive success across environments.

that positive selection is for thermal adaptation instead. Within COX1, 17 of 18 positively selected amino acid changes co-localize to proton translocation channels, with six occurring within four Å. These changes would be predicted to change proton pumping and our direct experimental assessment of proton pumping confirms a striking change relative to *C. elegans* (Fig. 5h). Combined with our data showing that gene expression changes are insufficient to explain the observed temperature-driven functional differences (Fig. 5i), the most plausible explanation is that positively selected amino acid changes contribute to a thermally tuned mitochondrial system. However, formal proof that the 20 FdNs directly alter mitochondrial respiration awaits mitochondrial gene sequence replacement methods in either *H. mephisto* or *C. elegans* or heterologous expression studies in other organisms such as yeast. These methods remain to be developed for metazoan organisms.

Our work also uncovers a novel type of regulation similar to engineered thermocouples, which are electrical devices designed to detect temperature differences and convert them into voltage[57,58]. In a thermocouple device, the difference in temperature between a hot and a cold junction produces an electrical voltage (Fig. 6f); in *H. mephisto* the voltage differences occur within animals distributed across the environment where they may encounter low temperatures (low voltage) or high temperatures (high voltage) (Fig. 6g). An ectothermic animal would gain significant advantage from efficiently coupling temperature to reproductive rates, thereby minimizing wasteful reproduction in non-optimal thermal environments. The direct linkage of mitochondrial respiration with temperature and lifecycle represents an elegant solution to this problem, a biologically evolved thermal-couple (Fig. 6i). Our work demonstrates an evolved adaptational metabolic regulation occurring most likely through amino acid changes to core metabolic machinery. This work foregrounds an underappreciated mode of evolutionary adaptation: the invention of a more responsive mapping of metabolism and reproduction onto environmental conditions.

## Methods
### Assembly of *H. mephisto* mitochondrial genome
As a component of a prior sequencing effort[2], PacBio RSII data were generated from raw genomic DNA. The data generated by three RSII lanes were assembled using HGAP3 and polished with Quiver (using the SMRT Analysis Software v2.3.0 pipeline), and the resultant assembly contained a contig encoding the complete mitochondrial genome (unitig_62|quiver). To validate the sequence, an Oxford Nanopore GridION low-coverage sequencing run of *H. mephisto* genomic DNA using a Flongle flow cell produced 6.5x average nuclear coverage and 47x coverage of the mitochondrial genome. After assembly using Flye 2.9.4, a single contig (28.6 kb) containing two complete copies of the mitochondrial sequence was recovered. This sequence aligned to the PacBio-derived sequence with 99.4 % identity (99.9 % identity for coding sequences), showing a few small indels around homopolymer runs characteristic of Nanopore sequencing error profile. The overall GC content of the Nanopore mitochondrial contig was 19.1 % while the GC content of the PacBio mitochondrial sequence was 19.0 %. Because no polishing was performed on this Nanopore sequence it was not deposited in a public repository.

### Curation of *H. mephisto* mitochondrial genome
The *H. mephisto* mitochondrial genome was aligned with that of *B. xylophilus* (AP017463.1), *P. redivivus* (AP017464.1), and *H. gingivalis* (KM192363.1) using MAFT v 7.017. Based on the clade IV mtDNA alignment and the results of MITOS WebServer v. 2.058, annotations were manually curated to represent all coding sequences, two ribosomal RNA molecules, and 22 transfer RNAs, and an AT-rich control region.

### Assembly of *Diploscapter pachys* mitochondrial genome
To ensure good sampling across the nematode phylogeny we selected at least six species from all five clades; we found that *D. pachys* mitochondrial data are not yet published. Upon inquiry the Gunsalus lab graciously shared this assembly. The *D. pachys* mitochondrial genome was assembled from Nanopore genomic reads by flye 2.8.1 and polished with Pilon 1.23. Pilon did not make any more corrections after iteration 3. Four indels were found by manual inspection of Illumina alignments, all of them in runs of Ts and within coding regions. These were then manually corrected to produce the current version of the *D. pachys* mitochondrial genome (13393 bps).

### Phylogenetic analysis
For the catenated protein tree, all coding sequences concatenated in the following gene order: COX1, COX2, ND3, ND5, ND6, ND4L, ND1, ATP6, ND2, CYTB, COX3, and ND4. Concatenated amino acid sequences were aligned with MAFT v.7.4.90, and the ML tree constructed using IQ-Tree59 1.6.12 webserver, allowing automatic detection of optimal substitution model (which for this dataset was mtZOA+F+I+G4). The Bayesian tree was constructed using Mrbayes 3.2.6 from the same alignment with the equalin amino acid rate matrix and the invgamma rate parameter.

### dN/dS analysis using PAML's branch-site model
In this method, individual branches are tested separately and a likelihood ratio test yields a *p* value for the probability positive selection has occurred on the lineage; for positive branches specific amino acid codons are identified as selection sites[37]. To correct for multiple hypothesis testing, a simple Bonferroni correction was performed for each gene. The nucleotide sequence for each protein-encoding gene was extracted and catenated together for each species. The resulting sequences were translation-aligned using Geneious Prime v.8.1.9 and all gaps were excised while maintaining coding frames of all sequences. Codeml of PAML v.4.9 was used to estimate branch site dN/dS, $\omega$ for sequence alignments for the following genes: COX1, COX2, COX3, and NADH4. The branching clades for PAML were provided as the IQ-Tree ML phylogenetic tree as described above. In Codeml, branch site $\omega$ was estimated using runmode = 0, seqtype = 1, model = 2, and NSsites = 2 in the control file. To enable likelihood ratio testing, every branch tested was run under two competing models: a neutral or relaxed selection model (fix_omega = 1) and a positive selection model (fix_omega = 0). *p* values were evaluated from the two likelihood values produced from the runs as follows. First, we generated a LRT statistic, $2\Delta l$. This statistic was compared against $\chi^2$ tables with two degree of freedom (df = 2). Thus, the critical value was set to 5.991 for 5% significance and 9.210 for 1% significance. To correct for multiple testing, we performed a

Bonferroni correction: the raw $p$ values multiplied by the number of tests run in a single tree. All PAML sites reported were identified by the Naive Empirical Bayes (NEB) analysis from the fix_omega = 0 runs, with a posterior probability of 0.95 or greater (a 5% significance level).

### Calculation of pairwise dS values
The Codeml package within PAML v. 4.9j was run in a pairwise mode as described[36] and in the user documentation. Specifically, a pairwise codon alignment was provided in phylip format as a user tree, and the runmode variable was set to −2 (pairwise), seqtype = 1 (codons), model = 1, and icode = 4 (for invertebrate mitochondrial genome translation table).

### Structural modeling
SWISS-MODEL was accessed via the Expasy web server, which identified bovine crystal structure 3abm as a suitable template for modeling. *H. mephisto* COX1, COX2, and COX3 were provided for modeling. The resultant model displayed GMQE of 0.82 and QMEANDisCo of 0.75 +/− 0.05. The model was visualized in PyMol v. 2.5.1 and was virtually superimposable (using command 'alignto 3abm [or 7coh], method=super') with two bovine structures 3abm (the template) and 7coh with RMSD of 0.142 (5880 of 5880 atoms) and 0.174 (5845 of 5845 atoms) Å, respectively. The H- and K- pathways were marked using the amino acid residues from ref. 59.

AlphaFold: ColabFold, an optimized AlphaFold 2 v. 2020 protein folding program, was accessed through ChimeraX v. 1.8rc202405220104 (2024-05-22) to generate multiple sequence alignment based models of *H. mephisto* COX1, COX2, and COX3. These structures were then superimposed with the SWISS-MODEL COX structure in PyMol v. 2.5.1 using the command "alignto Hmeph_COX123, method=super", with resulting RMSD values of 0.565, 1.592, and 0.815 respectively.

### Nematode culture, TMRE and sodium azide
The TMRE plates were prepared from standard, 60mm OP50-seeded NGM plates 61 supplemented with 500 $\mu$L of 4 $\mu$M TMRE dissolved in M9, evenly spread and allowed to soak into the plate and dry for 4–6 hours prior to adding nematodes. (Preparation of the 4 $\mu$M stock was by 1:1000 dilution from a 4mM TMRE-DMSO stock. A control DMSO plate was prepared and used in parallel to control for autofluorescence. All images of DMSO-only nematodes were blank so are not shown). Bleach-synchronized, starved L1 hatchling larvae of both *C. elegans* and *H. mephisto* were plated on NGM (no TMRE) and cultured at either 25 °C (*C. elegans*) or 37 °C (*H. mephisto*) for two days to reach adulthood. Worms were then washed to TMRE plates and cultured overnight at the temperature for which the assay is to be conducted (20 °C, 25 °C, or 37 °C), followed by destaining of TMRE for 1hr in 100$\mu$L M9, also at the assay temperature. Sodium azide (VWR Catalog # TS19038-0050) was used at 25mM final concentration and was added to appropriate samples during the 1hr TMRE destain period. All nematodes were paralyzed in a final concentration of 20mM Levamisole (AmBeed Cat # A121733-1G) immediately prior to imaging. For imaging, slides were made with 2% agarose pads in M9 and sealed with clear nail polish. Imaging was performed on an Olympus FV1200 scanning confocal microscope and 60x oil immersion objective (600x total magnification) focused on the body-wall muscle just behind the head area for each worm. Identical non-saturating laser settings were used on all images to ensure comparable, quantitative results, and images were processed using ImageJ v. 1.53t to determine the intensity of the spots per unit area. Statistical analysis was performed by two-way ANOVA and Tukey's HSD post-hoc test in R version 4.2.1.

### Seahorse Analysis of oxygen consumption rates
A Seahorse HS Mini device was used for all Oxygen Consumption Rate (OCR) analysis, following nematode protocols as previously described[20,21]. We found that the Seahorse XFp fluxPak (Agilent 103022-100) is essential- not the HS Mini plates, which have a ring structure on the bottom of the microchamber that often excludes worms from the measurements. The XFp plates have a characteristic 3 dots visible in the microchamber and work much better.

To measure maximal respiration we used Carbonyl cyanide-p-trifluoromethoxy-phenylhydrazone (FCCP), a potent decoupling agent of the inner mitochondrial membrane. We tested a variety of concentrations and found maximal response at 40 $\mu$M (final concentration) in both *C. elegans* and *H. mephisto*. Consistent with previous findings[21], we observed that high (1%) DMSO final concentration was required for FCCP function; whether this is due to improved drug solubility or absorption by the nematodes is unclear but lower DMSO (<0.1%) yielded no response. Thus all analyses were performed in 1% DMSO final concentration and we measured a 1% DMSO control. By using FCCP and sodium azide (final concentration 25 mM, soluble and functional in M9) we were able to measure basal, maximal, and spare capacity. Another compound, Dicyclohexylcarbodiimide (DCCD) has been shown to work in *C. elegans* to inhibit ATP synthase and thus to reveal ATP dependent oxygen consumption[21]; while we were able to replicate this, we found the drug had no effect on *H. mephisto* so we omitted it from our analysis.

Whenever possible we cultured the nematodes for three days at the Seahorse testing temperature (20 °C or 25 °C for *C. elegans* and 37 °C for *H. mephisto*). However, *H. mephisto* cultured at 20 °C, 25 °C, or 40 °C poses special challenges because of extremely slow growth at these suboptimal temperatures. Therefore, for these experiments we cultured *H. mephsto* for two days at 37 °C, enabling them to reach young adulthood, and then shifted to the testing temperature for an additional day prior to assay. We found that 11–13 *C. elegans* worms per well was optimal. Because *H. mephisto* are smaller we found that 60–80 worms per well was optimum. In all cases nematodes were visually counted under a light microscope before (*H. mephisto*) or before and after (*C. elegans*) a run. Because Seahorse analysis of nematodes is noisy, so we followed recommendations to allow many measurement cycles and take averages[21]. Therefore, one run consisted of 8 basal untreated measurements (prior to drug injection), 20 measurements after FCCP addition (or 1% DMSO control) and then 4 measurements after adding sodium azide addition. Each measurement cycle consisted of a 2 min mix, 2 min wait and 3 min measure period. We always performed 3 wells of FCCP injection and 3 wells of 1% DMSO injection, with two designated background wells.

Because DMSO had a strong effect on *H. mephisto* and *C. elegans* OCR (Fig. S4), we used the 1% DMSO as basal measures in Fig. 6c, d. (Untreated measurements for Fig. S4 were taken from measurements 5–8 prior to any drug injections). We found that optimal measures for maximal responses were different between species, with *H. mephisto* peaking and dropping off more dramatically than *C. elegans*. Thus, we used four measurements, 10–13 for *H. mephisto*, whether of 1% DMSO controls (which we count as basal respiration in Fig. 6c) or FCCP (for maximal respiration). For *C. elegans* we used measurements 13–16, in accord with Luz et al.[21], for the FCCP or 1% DMSO basal measure (Fig. 6d).

Once we had obtained the data from the runs, we normalized OCR to pmol/min/worm and performed the following calculations following Agilent Seahorse guidelines: basal is 1% DMSO OCR minus the non-mitochondrial oxygen consumption (OCR after sodium azide). Maximal respiration is FCCP OCR minus non-mitochondrial OCR (sodium azide). Spare capacity is FCCP OCR minus 1% DMSO OCR.

### RNAseq data analysis for mitochondrial expression
Previously sequenced RNA-seq data 2 were mapped to the mitochondrial genome with HISAT2 2.2.1 and Stringtie 2.2.1 and Ballgown 2.26.0 installed and run in R 4.1.0. Differential expression was assessed with the stattest() function of Ballgown.

### Hypoxia culture and survival assessment
Nematodes were grown in anaerobic chambers. We seeded NGM plates with synchronized *C. elegans* or *H. mephisto* L1 larvae from overnight hatching in M9 (at 20 °C for *C. elegans* and 37 °C for *H. mephisto*), allowing a few minutes to dry, and then placing the plates into 2L chambers (Cat # 260002) with BD GasPak EZ anaerobe container system sachets (Cat # 260678) to create an anaerobic environment. Nematodes were cultured at

either 20 °C or 37 °C depending on the species in standard incubators, for the indicated time (2 or 9 days) before opening and measuring viability. Survival was counted by washing the worms off the plates and exposing them to Sytox Orange dye (Thermo Fisher Cat # S11368), diluted 1:1000 in M9, for 15 minutes. Worms were then collected by centrifugation (400 g for 2 minutes), pipetted onto 2% agarose-M9 pads, coverslipped, and sealed with clear nail polish prior to measurement of the dead animals using an Olympus BX61 fluorescence microscope set to the TRITC channel. Worm lengths (live animals only) were measured at the same time using CellSense v. 2.3 software.

### Statistics and reproducibility

Statistical analysis was perfored by one or two way ANOVA followed by Tukey's HSD post-hoc test or a two-tailed Students T-test of unequal variance, all implemented in R version 4.2.1.

### Data availability

The mitochondrial sequence and annotation of *H. mephisto* has been deposited to GenBank under accession OP965539. For all figures showing numeric results, the source data are provided in the supplemental data 1 file. All other data available from the corresponding author upon request.

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

## Acknowledgements
We acknowledge Dr. Stefano Costanzi for his guidance and suggestions regarding 3D structural modeling and visualization. We also thank our anonymous reviewers for their feedback. We also acknowledge NIH grant 1R15GM146207 to J.R.B.

## Author contributions
Megan Guerin performed formal analysis, visualization, and writing-original draft. TreVaughn Ellis performed formal analysis, visualization, investigation, and conceptualization. Mark Ware performed formal analysis and visualization. Alexandra Manning performed investigation. Ariana Coley performed investigation, formal analysis, visualization, and conceptualization. Ali Amini performed formal analysis, visualization, software, and conceptualization. Adaeze Igboanugo performed investigation. Amaya Rothrock performed investigation. George Chung performed investigation. Kristin Gunsalus performed conceptualization. John Bracht performed conceptualization, visualization, formal analysis, investigation, funding acquisition, supervision, project administration, and writing-review and editing.

## Competing interests
The authors declare no competing interests.
