## [Peer review file · Communications Biology]

Reviewers' comments:

Reviewer #1 (Remarks to the Author):

1. What are the major claims of the paper? Are they novel and will they be of interest to others in the community and the wider field?

In this paper, an extremophile/ thermophile metazoan, *Halicephalobus mephisto* was discovered to have a thermal-regulated cytochrome C oxidase that loses activity at low temperatures (20 °C) but recovers and has optimal activity at a higher temperature (37 °C). This affects the organism's growth rate in a corresponding manner and allows the organism to regulate itself without expending energy.

The conclusions are novel. I was especially impressed by the results that suggested the amino acid differences in COX1 were for thermal and not hypoxia adaptation. I do think this will be of interest to others that study metazoans and thermal adaptation.

2. Is the work convincing, and if not, what further evidence would be required to strengthen the conclusions?

This work is convincing to me. I think the authors have done a nice job analyzing the COX genes and proteins in a novel nematode with an unusual home environment.

3. On a more subjective note, do you feel that the paper will influence thinking in the field?

I do believe this paper will influence thinking in the field of electron transport/ cytochrome C oxidase and will also influence thinking about thermal adaptation. The finding that evolutionary pressures on this organism would cause amino acid substitutions in COX to influence thermal adaptation and not hypoxia, is a clear and novel result. While mitochondrial and metazoan adaptation is not my field, I find these results convincing. This will be something that other researchers will be looking for in other thermally adapted organisms.

4. Other questions/ comments.

The introduction is very short. I would expand your literature review on the organism. Some suggestions: why it is of interest, other related members of its family, etc., adaptations in cytochrome C oxidase and the electron transport chain. You may also want to include a brief survey of other nematodes/metazoans (and their cytochrome C oxidase/ electron transport) and what we know about the mesophilic versus the thermophilic ones (if any). This background literature will strengthen your research and further bolster the unique study that it is.

I am intrigued by the really large percentage of AT's in the mitochondrial genome. This is contrary to most modestly thermally adapted organisms, which have higher GC/ lower

AT content. It would be good to confirm this result on another sequencing platform. Because related nematodes have a similar percentage AT, it would be very interesting to explore how mitochondria in this organism deal with the heat without possible thermal effects/ melting. Could it be that the really high AT-rich control region melts and provides a trigger for some cellular process?

Creating a structural homology model of the *H. mephista* cytochrome C oxidase was a good way to visualize the substitutions and SWISS-MODEL is an acceptable software to do this. But, I have doubts that the model it created here is an accurate representation of the *H. mephista* COX. Those RMSD values (0.142 and 0.174) are too good. There are differences between the cow and *H. mephista* COX sequences -including variety (polar, nonpolar, etc.) and number of amino acids (514 vs 524, using a quick search)- and this must impact the structure in some way. You have two choices here: you can state that this is a model for illustrative purposes only (which is fair and acceptable) or you can repeat the model building with other software, like CHIMERA X (<https://www.cgl.ucsf.edu/chimerax/>). CHIMERA X gives good results (with energy minimization and other onboard tools) that are more believable with proteins that are similar.

I'm not sure why information about the *Diploscapter pachys* mitochondrial genome was included in the Materials and Methods section. This paper did not talk about this organism very much. A short explanation would reduce confusion. Was this organism sequenced to provide another point for the phylogenetic analysis?

5. Typo

In the Methods section (p. 18, line 602), it is MrBayes, not Mr. Baye's. Unless Mr. Baye's is a different software package.

Reviewer #2 (Remarks to the Author):

There are grammatical issues throughout the manuscript. I have not commented on each grammatical issue however please note that the manuscript needs detailed editing before publication.

The manuscript is framed around the mitochondrial assembly of *H. mephisto*. The reads are taken from another project, which makes me wonder why the mitochondrial assembly was not included with a nuclear genome assembly for this species. I recommend re-framing the manuscript around the interesting thermal findings (which need additional studies for robustness) instead of the mitochondrial assembly. This

seems to be a mitochondrial genome assembly report, but the more interesting part is the physiological studies. However, I am not a physiologist so I cannot determine whether the detailed methods are appropriate.

Regarding the molecular evolution analyses: The outgroups are way too far for studies of molecular evolution, likely have codon site saturation. This is mentioned later in the manuscript and other approaches are utilized (fixed, derived nonsynonymous substitutions) which are going to misrepresent the amount of selection. The BEB results from PAML branch-sites test not used to determine the sites of interest because of site saturation. The paragraph starting line 121 defines the non-synonymous sites but the branch-sites test would have identified the sites putatively under positive selection. It is also unclear whether a model with $w=1$ was tested in the branch-sites context. It turned out this information was in the following paragraph but the results need to be re-written for clarity. I highlight that if there is synonymous site saturation, then a smaller subset of the tree should be considered for the dn/ds analyses and then the robust method (BEB) could be used instead of just choosing derived fixed nonsynonymous substitutions.

The conclusion on line 174, that because the worms are sensitive to hypoxia at high temperature means that the amino acid substitutions must be temperature adapted does not make sense.

It is necessary to test additional temperatures beyond just two temperatures, especially because *C. elegans* was only tested at one temperature. Higher temperatures for *H. mephisto* would be informative for the phenotype testing.

Smaller points:

The first sentence is misleading because it sounds as though the manuscript is about experimental evolution.

Text such as “statistically robust evidence” is unnecessary.

Unclear how reproductive success is maximized across temperatures.

Mention which mitochondrial gene is lost in nematodes (line 56)

Confused about the addition of the *H. gingivalis* and *Reticulitermes speratus* genomes as something different from the 35 that were also added for the phylogenetic analysis.

The results and methods are confounded and need to be more clearly separated. There are too many methods in the results section.

We wish to thank both reviewers for their helpful comments. We have addressed them as discussed below and we are pleased with the way this has strengthened the manuscript.

Reviewer #1:

1. What are the major claims of the paper? Are they novel and will they be of interest to others in the community and the wider field?

In this paper, an extremophile/ thermophile metazoan, *Halicephalobus mephisto* was discovered to have a thermal-regulated cytochrome C oxidase that loses activity at low temperatures (20 oC) but recovers and has optimal activity at a higher temperature (37 oC). This affects the organism's growth rate in a corresponding manner and allows the organism to regulate itself without expending energy. hacid differences in COX1 were for thermal and not hypoxia adaptation. I do think this will be of interest to others that study metazoans and thermal adaptation.

We appreciate the reviewer's comment.

2. Is the work convincing, and if not, what further evidence would be required to strengthen the conclusions?

This work is convincing to me. I think the authors have done a nice job analyzing the COX genes and proteins in a novel nematode with an unusual home environment.

We thank the reviewer for the feedback.

3. On a more subjective note, do you feel that the paper will influence thinking in the field?

I do believe this paper will influence thinking in the field of electron transport/ cytochrome C Oxidase and will also influence thinking about thermal adaptation. The finding that evolutionary pressures on this organism would cause amino acid substitutions in COX to influence thermal adaptation and not hypoxia, is a clear and novel result. While mitochondrial and metazoan adaptation is not my field, I find these results convincing. This will be something that other researchers will be looking for in other thermally adapted organisms.

We again thank the reviewer.

4. Other questions/ comments.

The introduction is very short. I would expand your literature review on the organism. Some Suggestions: why it is of interest, other related members of its family, etc., adaptations in cytochrome C oxidase and the electron transport chain. You may also want to include a brief survey of other nematodes/metazoans (and their cytochrome C oxidase/ electron transport) and what we know about the mesophilic versus the thermophilic ones (if any). This background literature will strengthen your research and further bolster the unique study that it is.

We thank the reviewer for the comment. We agree the introduction needs more detail and have added a paragraph discussing the study of mitochondria from thermally-adapted animals. Most studies simply report sequence changes (without functional studies) or perform functional analysis of mesophilic animal mitochondria. In our work we combine functional studies with a stress-adapted organism, so our study occupies a unique position in the literature. We think this additional context is important to frame our findings relative to the overall literature.

I am intrigued by the really large percentage of AT's in the mitochondrial genome. This is contrary to most modestly thermally adapted organisms, which have higher GC/ lower AT content. It would be good to confirm this result on another sequencing platform.

We appreciate the suggestion. Recently the PI acquired an Oxford Nanopore GridION sequencer so we used this in-house resource to confirm the PacBio sequencing result. Genomic *H. mephisto* DNA was sequenced on a small flowcell (Flongle) to 6.5x coverage of the nuclear genome and 47x coverage of the mitochondrial genome. This data was assembled using Flye yielding an assembly of 1,749 contigs, one of which contained a circular concatemer of the entire mitochondrial assembly (a 28.5 kb contig). We broke this contig into two complete mitochondrial genomes and aligned them to the PacBio mitochondrial sequence. The alignment quality was excellent, at 99.4% identity (99.9% in coding sequences) relative to PacBio. The GC content was nearly identical: 19.1% GC for the Nanopore sequence vs. 19.0% GC for PacBio. Nanopore variants relative to PacBio occur predominantly at 18 regions, mostly homopolymers and enriched for intergenic sequences, and some of these variants were corrected in one of the Nanopore sequences versus the other (since the contig recovered two full mitochondrial genomes). This suggests they are predominantly sequencing errors. The variations we were seeing are consistent with the known error profile of Nanopore sequencing which is higher at homopolymeric sequences. Given that we performed no polishing of the Nanopore assembly, we consider the PacBio data validated and we have added this to the Methods section of the manuscript. Because this Nanopore contig is unpolished we did not deposit it into GenBank or make it publicly available.

Because related nematodes have a similar percentage AT, it would be very interesting to explore how mitochondria in this organism deal with the heat without possible thermal effects/ melting. Could it be that the really high AT-rich control region melts and provides a trigger for some cellular process?

We find this suggestion very intriguing. While it is true that some bacteria tend to have higher GC content if they are more thermally adapted, the trend is less clear for archaea and eukaryotes. Furthermore, while *H. mephisto* grows at high temperature for a nematode, it is not a hyperthermophile with its growth optimum (37°C) being human body temperature and the growth maximum of 40°C in the laboratory. The 100% AT-region in the control region of the genome (265 bp) is predicted to have a T_m of 66°C, much higher than the culture maximum of *H. mephisto*. Therefore we would not expect even AT-rich DNA to melt appreciably into single

strands unless facilitated by DNA helicases which is something to be investigated in a future study.

Creating a structural homology model of the *H. mephisto* cytochrome C oxidase was a good way to visualize the substitutions and SWISS-MODEL is an acceptable software to do this. But, I have doubts that the model it created here is an accurate representation of the *H. mephisto* COX. Those RMSD values (0.142 and 0.174) are too good. There are differences between the cow and *H. mephisto* COX sequences -including variety (polar, nonpolar, etc.) and number of amino acids (514 vs 524, using a quick search)- and this must impact the structure in some way. You have two choices here: you can state that this is a model for illustrative purposes only (which is fair and acceptable) or you can repeat the model building with other software, like CHIMERA X (<https://www.cgl.ucsf.edu/chimerax/>). CHIMERA X gives good results (with energy minimization and other onboard tools) that are more believable with proteins that are similar.

We appreciate the reviewer's comment. We have reworked how we report the numbers. Reporting RMSD of a structure relative to its template (in this case, bovine crystal structure 3abm) will of course show a good match since the structure is built off that template. Therefore, the 0.142 is not really useful (except to show the homology prediction algorithm worked as it was supposed to). Therefore we have moved these data to the methods section, and we decided to use AlphaFold as an independent approach. When we performed the structural analysis reported in our paper AlphaFold, wasn't available, so this was a great opportunity to check our work.

We have now run AlphaFold and generated homology models for *H. mephisto* COX1, 2, and 3. The match of the SWISS-MODEL structure to the AlphaFold structures was quite good, especially for COX1, as shown in the following table. We consulted a protein structure expert within the Chemistry department of our university (Dr. Stefano Costanzi) who considered AlphaFold to be nearly as good as crystal structures in most cases. Thus these results support the accuracy of the original homology structure we generated.

Gene	RMSD to H. mephisto SWISS-MODEL
COX1-alphaFold	0.565
COX2-alphaFold	1.592
COX3-alphaFold	0.815

[Note: *H. mephisto* Swiss-Model has COX1, 2, and 3 combined in one structural prediction.]

We have created an additional supplemental figure showing the overlay of the AlphaFold with the Swiss-Model version (Figure S2). Given the structural similarity and the fact that the structure is meant as a hypothesis-generating illustration (as suggested by the reviewer), we left the Swiss-Model as the main figure. An additional advantage is that the SWISS-MODEL structure, being built off a multi-subunit template structure, includes the relative positioning of COX1, 2, and 3 relative to each other, while the AlphaFold structures are separate. This makes

the SWISS-MODEL structure more comprehensive and complete, as extra docking or would be required to show the AlphaFold structures together. We have adjusted the text accordingly to indicate the purpose of the model is to be illustrative only.

I'm not sure why information about the *Diploscapter pachys* mitochondrial genome was included in the Materials and Methods section. This paper did not talk about this organism very much. A short explanation would reduce confusion. Was this organism sequenced to provide another point for the phylogenetic analysis?

Yes, we felt that *Diploscapter* would add valuable phylogenetic information. We made a conscious effort to sample broadly across nematode phylogeny when we constructed our sequence alignment and phylogenetic tree. When we analyzed the *D. pachys* genome we found the mitochondrial sequence was not there; we reached out to the Gunsalus lab who performed special efforts to assemble, polish, and annotate this sequence. We have added some text to the manuscript to clarify this.

5. Typo

In the Methods section (p. 18, line 602), it is MrBayes, not Mr. Baye's. Unless Mr. Baye's is a different software package.

It has been corrected. We thank the reviewer for pointing this out.

Reviewer #2:

There are grammatical issues throughout the manuscript. I have not commented on each grammatical issue however please note that the manuscript needs detailed editing before publication.

We thank the reviewer and we have corrected the grammatical issues throughout the text.

The manuscript is framed around the mitochondrial assembly of *H. mephisto*. The reads are taken from another project, which makes me wonder why the mitochondrial assembly was not included with a nuclear genome assembly for this species. I recommend re-framing the manuscript around the interesting thermal findings (which need additional studies for robustness) instead of the mitochondrial assembly. This seems to be a mitochondrial genome assembly report, but the more interesting part is the physiological studies. However, I am not a physiologist so I cannot determine whether the detailed methods are appropriate.

A nuclear genome and a mitochondrial genome are very different and warrant individual attention. When we published the nuclear genome assembly from *H. mephisto* we performed comparative and functional analysis. We chose a similar route for the mitochondrial genome in this project. We believe this makes a more compelling study than just a "parts list" of the mitochondria. The additional studies suggested by the reviewer are included in this revised manuscript.

Regarding the molecular evolution analyses: The outgroups are way too far for studies of molecular evolution, likely have codon site saturation. This is mentioned later in the manuscript and other approaches are utilized (fixed, derived nonsynonymous substitutions) which are going to misrepresent the amount of selection. The BEB results from PAML branch-sites test not used to determine the sites of interest because of site saturation. The paragraph starting line 121 define the non-synonymous sites but the branch-sites test would have identified the sites putatively under positive selection. It is also unclear whether a model with $w=1$ was tested in the branch-sites context. It turned out this information was in the following paragraph but the results need to be rewritten for clarity. I highlight that if there is synonymous site saturation, then a smaller subset of the tree should be considered for the dn/ds analyses and then the robust method (BEB) could be used instead of just choosing derived fixed nonsynonymous substitutions.

We appreciate these concerns from the reviewer. We shared many of the same concerns about the high divergence within the phylogeny which is why we didn't rely solely on PAML and its identification of sites and instead used fixed derived non-synonymous (FdNs) substitutions as a better measure of positive selection. We wish to make two points:

First, The branch-sites method implemented through PAML has been extensively characterized including cases where synonymous mutations (dS) are high (saturating)¹⁻⁴. While a different method, pairwise dN/dS comparison (i.e., an analysis using two different sequences only), has been shown to produce artifactually high estimates of ω when synonymous mutations (dS) are saturating⁵, the tree-based branch-sites method used in our study (with 38 total sequences) optimizes rate parameters across a mathematical model, so it is not prone to this error. Instead, the branch-sites method produces a loss of power (false negatives) rather than false positives under saturating dS⁴. So methodologically this is less of a concern. Indeed, the fact that a method lacking in power at high dS has still managed to pick out the same sites as the manual inspection of alignments (see below) suggests that there is real signal in the data.

Second, we prioritized the analysis of Fixed, Derived Non-Synonymous Substitutions (FdNs) which can be obtained simply by inspecting an alignment of the coding sequences, without PAML's involvement. The FdNs method is essentially a tracking of amino acid replacements within a clade and is regularly used. For example, adaptive evolution in woolly mammoth⁶, a pioneering paper to track positive selection in COX⁷ and cytochrome c⁸ in the anthropoid primate lineage. (Note: FdNs involve more than just looking for amino acid replacement, it requires that the observed amino acid changes be correlated to the same nucleotide change at the DNA level. However, to a first approximation one can track amino acid replacements across a phylogeny and many papers stop at that level. We go one step beyond to the FdNs level which is more accurate.)

The reviewer seems to disparage FdNs analysis as somehow misrepresenting the "amount of selection". We dispute this claim, which was offered with no justification by the reviewer. Indeed, FdNs substitution analysis captures the key element of adaptive evolution, which is the retention of selectively advantageous alleles. In contrast, the "amount of selection" appears to be a vague

and unquantified term, though we suspect the reviewer meant something like relative selective advantage or selection strength. However, FdNs analysis truly captures what evolution does. It encapsulates the outcome of natural selection on allelic variation which is what matters—regardless of selective strength, "amount of selection" or any other estimator. We would argue FdNs analysis is the superior metric.

Under conditions of high evolutionary divergence, with high synonymous substitution rates, the retention of beneficial alleles just by chance is highly unlikely—as evolution explores neutral space, the conserved variants should mutate to other bases. Indeed in our alignment we could identify 8 of 20 total FdNs *within codons that themselves contain synonymous substitutions across the alignment*. In other words, it appears as if evolution is retaining the FdNs substitutions, against a high background of synonymous mutations among the clades under study. This means that once it has occurred, the FdNs spreads (presumably under $dN/dS > 1$) and then becomes a site of purifying selection again ($dN/dS < 1$). Or, to put it another way, positive selection leads to purifying selection after the spread of adaptive alleles, an accepted model of evolutionary change⁹. We disagree with the reviewer that this "misrepresents the amount of selection". This method accurately captures the key adaptive events in evolutionary history and even captures the way in which positive selection leads to purifying selection⁹.

We also wish to highlight that the FdNs and PAML methods produced overlapping mutually supportive results. The PAML results inevitably included a few more sites than the FdNs analysis which thus appears to be more conservative. We carefully explained all this in the manuscript to make clear the relative outputs of each method. We believe this careful approach (and using several methods that are mutually reinforcing) is suitable to the present study. If the analysis of amino acid replacement across phylogenies is not valid, then entire subfields within evolutionary genomics must be discarded along with thousands of published papers.

The reviewer is correct that the sites we discuss in the paper were Naive Empirical Bayes (NEB) sites (all with a posterior probability 0.95 or greater). BEB analysis produced no statistically significant outputs to posterior probability of 0.95 or higher. To address this, the reviewer suggests using a smaller subset of the tree for PAML analysis to obtain better BEB results. We weren't sure whether this would make a difference for a theoretical reason: the branch-sites analysis with PAML requires pre-specification of one or more clades as "foreground" lineages being tested for positive selection while other clades are "background" lineages. Thus doing as the reviewer suggests (using a smaller overall tree) seems most likely to result in lower statistical power as it removes species from the background. (We do not think the reviewer is asking us to test removal of foreground lineages as that would increase the evolutionary space between remaining species on the tree.) So, assuming all foreground lineages remain the same, the foreground dS values won't be altered—whether selected from a large tree (38 species) or smaller sub-tree. Given a smaller sub-tree has fewer background clades we would hypothesize that lower statistical power of results would occur.

To test this we performed a subset analysis using sub-trees of 1) clades IV and V only (Tylenchina and Rhabditina), a total of 14 species, and 2) a minimal subset of 8 species of clade

IV only (Tylenchina) rooted on the *Steinernema* species and excluding *Diploscapter*. (For this analysis, all phylogenetic trees were re--constructed from protein alignments with IQ-Tree and branches A, B, C, and D were tested as foreground lineages in PAML branch-sites test).

As noted above, no statistically significant BEB sites were found in the PAML outputs (just NEB sites). However, a few (not significant to posterior probability 0.95) BEB sites were identified in the PAML outputs and are listed below. Consistent with a hypothesis of a loss of power, there were fewer BEB sites identified in subset phylogenies, and they had lower posterior probabilities from smaller trees:

(Position = amino acid alignment position; square brackets are posterior probabilities given by PAML).

Branch Number	Original (38 species) position [posterior prob]	Subset, 14 species position [posterior prob]	Subset, 8 species position [posterior prob]
A	318 [0.899],431[0.547]	318 [0.883]	318 [0.785]
B	461 [0.857]	none	none
C	486 [0.692]	none	none
D	none	none	none

It is clear that subset trees produce lower statistical power, not greater. Given this, and given our careful application of evolutionary analysis methods, we conclude that the PAML analysis provided in the manuscript is robust, reliable, and the reporting of BEB sites in PAML is not improved by analysis of subsets of the data as suggested by the reviewer.

The conclusion on line 174, that because the worms are sensitive to hypoxia at high temperature means that the amino acid substitutions must be temperature adapted does not make sense.

We have clarified the sentence. The point was to say that adaptation occurred (the FdNs and PAML outputs give evidence) and so we tested hypoxia. That wasn't the case, so we thought thermal adaptation was a better explanation for adaptation here.

It is necessary to test additional temperatures beyond just two temperatures, especially because *Celegans* was only tested at one temperature. Higher temperatures for *H. mephisto* would be informative for the phenotype testing.

We appreciate the comment from the reviewer. To expand the metabolic analysis we performed Seahorse analysis for *H. mephisto* at 25 °C and 40 °C, which is their maximum survivable temperature in the lab. This gives us a good match to both Figure 5 (where we measured 25 °C proton pumping) and also lets us see how they respond to a temperature that is stressful for them (40 °C). This expands the tested temperature to four total: 20, 25, 37, and 40 °C.

We also analyzed *C. elegans* at 25 °C, so we have data from 20 and 25 °C conditions. Together these data enrich our analysis and uncover some surprising patterns which we discuss in the text. We appreciate the opportunity presented by the reviewer to deepen our analysis.

Smaller points:

The first sentence is misleading because it sounds as though the manuscript is about experimental evolution.

We appreciate the comment and we changed "evolved" to "biological" in the first sentence of the abstract to make it more clear.

Text such as "statistically robust evidence" is unnecessary.

We removed this phrase from the abstract.

Unclear how reproductive success is maximized across temperatures.

We appreciate the comment because it reveals that we had inadvertently left unstated the link between individual life cycle time and population growth rates. We have clarified the text.

Mention which mitochondrial gene is lost in nematodes (line 56)

We appreciate the comment. The lost gene is ATP8. We have added this to the text.

Confused about the addition of the *H. gingivalis* and *Reticulitermes speratus* genomes as something different from the 35 that were also added for the phylogenetic analysis.

We apologize for the confusion and have corrected the text.

The results and methods are confounded and need to be more clearly separated. There are too many methods in the results section.

We have moved methods from the results section.

1. Yang, Z. PAML 4: phylogenetic analysis by maximum likelihood. *Mol. Biol. Evol.* **24**, 1586–1591 (2007).
2. Zhang, J., Nielsen, R. & Yang, Z. Evaluation of an improved branch-site likelihood method for detecting positive selection at the molecular level. *Mol. Biol. Evol.* **22**, 2472–2479

(2005).

3. Yang, Z. & dos Reis, M. Statistical properties of the branch-site test of positive selection. *Mol. Biol. Evol.* **28**, 1217–1228 (2011).
4. Gharib, W. H. & Robinson-Rechavi, M. The branch-site test of positive selection is surprisingly robust but lacks power under synonymous substitution saturation and variation in GC. *Mol. Biol. Evol.* **30**, 1675–1686 (2013).
5. Cannarozzi, G. M. & Schneider, A. *Codon Evolution: Mechanisms and Models*. (Oxford University Press, 2012).
6. Díez-Del-Molino, D. *et al.* Genomics of adaptive evolution in the woolly mammoth. *Curr. Biol.* (2023) doi:10.1016/j.cub.2023.03.084.
7. Schmidt, T. R. *et al.* Rapid electrostatic evolution at the binding site for cytochrome c on cytochrome c oxidase in anthropoid primates. *Proc. Natl. Acad. Sci. U. S. A.* **102**, 6379–6384 (2005).
8. Pierron, D. *et al.* Silencing, positive selection and parallel evolution: busy history of primate cytochromes C. *PLoS One* **6**, e26269 (2011).
9. Goodman, M. Positive selection causes purifying selection. *Nature* **295**, 630 (1982).

REVIEWERS' COMMENTS:

Reviewer #1 (Remarks to the Author):

All of my concerns have been satisfied. I have no additional concerns. The final manuscript looks good and will contribute to our understanding of thermal adaptation in nematodes.

Reviewer #2 (Remarks to the Author):

The authors have addressed my concerns.